# How leaders are persuaded: An elaboration likelihood model of voice endorsement

Xiaobo Li[1,2], Ting Wu[2]*, Jianhong Ma[1]

**1** Department of Psychology and Behavioral Science, Zhejiang University, Hangzhou, China, **2** School of Business, Zhejiang University City College, Hangzhou, China

* tingwu@outlook.com

## Abstract

Organizations need both employee voice and managerial endorsement to ensure high-quality decision-making and achieve organizational effectiveness. However, a preponderance of voice research focuses on employee voice with little attention paid to voice endorsement. Building on the social persuasion theory of the elaboration likelihood model, we systematically examine the sender and receiver determinants of voice endorsement and how the interplay of those determinants affects voice endorsement. By empirically analyzing 168 paired samples, we find that issue-relevant information, i.e., voicer credibility, has a positive effect on voice endorsement and matters most when leaders have high felt obligation. The results also show that the peripheral cue used in the study, i.e., positive mood, has a positive effect on voice endorsement and matters most when leaders have low felt obligation or low cognitive flexibility. We discuss the contributions of these findings and highlight limitations and directions for future research.

## Introduction

As the business environment becomes increasingly complicated and turbulent, frontline employees are uniquely positioned to address changing market demand, cutting-edge technological developments, and unexpected organizational issues [1]. Therefore, leaders in organizations need to encourage their employees to speak up to ensure high-quality decision-making and achieve organizational effectiveness [e.g., 2–5]. However, for these benefits to materialize, leaders need to endorse, accept, or positively receive the voiced advice [6–8]. In reality, leaders are tempted to dismiss employee voice because they think that endorsing employee voice will destroy the sets of organizational routines or authority they have created [9]. Hence, the effectiveness of voice involves the upward communication of employee voice that is intended to benefit the organization [3, 10] and the downward process of voice endorsement through which leaders are persuaded by their employees [3, 4].

Although voice endorsement is important, empirical research is limited. Among research studies, researchers mainly focused on how message factors influence voice endorsement and mostly ignored sender and receiver factors. For example, Burris (2012) found that leader voice endorsement of employees depends on the type of voice exhibited by those employees [7].

**Data Availability Statement:** All relevant data are within the manuscript and its Supporting Information files.

**Funding:** This work was supported by Hangzhou Social Science Project for Developing High Calibre Youth Talent under Grant No. 2018RCZX22,

Zhejiang Provincial Natural Science Foundation of China under Grant No. LQ18G020007, and Research Center of Digital Transformation and Social Responsibility Management, ZUCC.

**Competing interests:** The authors have declared that no competing interests exist.

Employees engaging in more supportive forms of voice are more likely to be endorsed than those who engage in challenging forms of voice. As another example, Lam et al. (2019) found that managers are more likely to accept and act on a direct voice with explicit change suggestions than on an indirect voice with hints [3]. In addition to the influence of the message, the voice sender and receiver may also affect voice endorsement [7]. In a recent study, Li et al. (2019) showed that employee voice was more likely to be endorsed by managers with low ego depletion than by managers with high ego depletion [11]. However, we still need more systematic evidence to deepen the understanding of how the interplay between the voice sender and receiver affects voice endorsement.

Current research examining the determinants of voice endorsement is primarily concerned with issue-relevant information, including the quality of the voice message [12] and the expertise of the voice sender [13]. However, as a decision process, voice endorsement is also influenced by peripheral cues, such as voice directness and voice politeness [3]. Both issue-relevant information and peripheral cues influence voice endorsement. According to Schreurs et al. (2020), voice endorsement is a complicated process that needs an integrative way of examining its determinants [4]. However, existing research is mainly concentrated on issue-relevant information and ignores peripheral cues. Therefore, the question that has not been fully answered is whether voice endorsement is influenced by some peripheral cues presented by employees and, if so, whether the influence process is the same as that used for issue-relevant information.

To address these research gaps, we first draw upon the social persuasion theory of the elaboration likelihood model (ELM) to examine how two different types of cues, issue-relevant information about voicer credibility and the peripheral cue of positive mood, influence voice endorsement. According to the ELM [14, 15], we propose that employee voicer credibility may trigger the central route through which leaders evaluate voice using critical thought, whereas positive employee mood, on the other side, may guide their leaders to participate in the peripheral route through which leaders are more likely to assess voice using affective states. Another goal of this study is to deepen the understanding of voice endorsement by investigating the interactive effects of sender and receiver factors. Specifically, the ELM indicates that the extent to which individuals choose to analyze the message provided by each route depends on their motivation and ability to evaluate the merits of a focal issue in a particular decision context [16]. Based on this argument, we propose that the central and peripheral routes of leader voice endorsement are influenced by a leader's felt obligation (motivation) and cognitive flexibility (ability). In general, we mainly study two research questions: (1) Will the voicer credibility and positive mood of employees influence voice endorsement by their leaders? And (2) How does the interplay of sender factors (voicer credibility and positive mood) and receiver factors (felt obligation and cognitive flexibility) affect voice endorsement?

The overall research model is shown in Fig 1.

## Literature

### The central route: Voicer credibility and voice endorsement

The significance of source credibility has not gone unnoticed by scholars of social persuasion, and many studies have linked voicer credibility to voice endorsement. For instance, Whiting et al. (2012) found that the voice put forward by a more credible employee is considered more constructive by his or her leader, resulting in better performance evaluation of that employee [17]. A recent study conducted in the context of entrepreneurship [15] provides indirect but valuable evidence for the impact of positive mood on voice endorsement. In that study, researchers found that when crowdfunding entrepreneurs present entrepreneur-specific

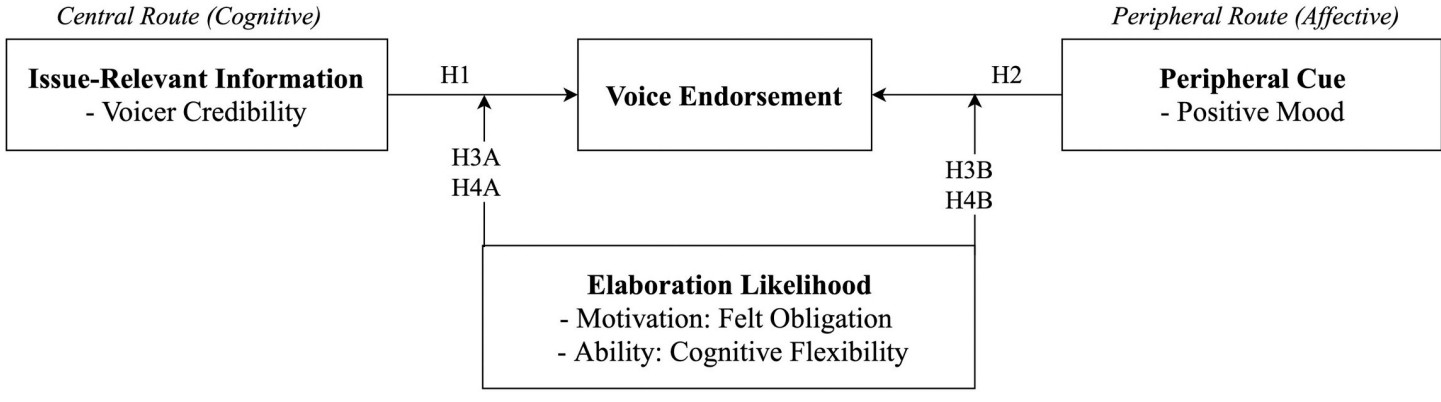

**Fig 1. An elaboration likelihood model of voice endorsement.**

information (e.g., formal education or work experience) that makes them credible, funders are more likely to be persuaded to provide capital.

From the perspective of the ELM, we expect an employee's voicer credibility will impact his or her leader's voice endorsement through the central route. There are two reasons for this. First, employee voicer credibility can foster a leader's ability to engage in issue-relevant thinking about the arguments in a focal message. As market environments, technological developments, and organizational issues become increasingly complicated, the "unfamiliar or unknown areas" for leaders gradually expand, limiting their ability to scrutinize issue-relevant information. However, voicer credibility, a signal that an employee has the expert knowledge necessary to make credible suggestions in a topic-specific field [17], can help leaders form new cognitive rules to deal with changing and complicated situations. Second, voicer credibility can increase the willingness of leaders to participate in more cognitive efforts. Because expert knowledge facilitates accurate problem recognition, voicer credibility also signals which employees deserve attention. As suggested by Whiting et al. (2012), message receivers have more positive attitudes toward the message when it is delivered by trustworthy experts [17]. Thus, we hypothesize the following:

**H1:** Voicer credibility is positively associated with voice endorsement.

### The peripheral route: Positive mood and voice endorsement

The role of mood has been considered one of the 16 critical issues in decision-making [18]. In prior research, most scholars believed that the influence of mood could be transferred only from high-power to low-power individuals rather than the other way around. That belief led to extensive efforts to determine the role of leader mood in upward communication of employee voice [e.g., 19, 20] and little attention to employee mood in downward communication of voice endorsement. However, a recent study on entrepreneurship tells a different story and reveals that the mood transmission can also proceed from low-power entrepreneurs to high-power funders.

Taking the ELM perspective, we assume that positive employee mood, as a peripheral cue, may affect leader voice endorsement through the peripheral route. According to Bhullar (2012), when exposed to positive framing, such as positive employee mood, leaders may "mirror" or mimic the mood of their employees [21]. Hence, by presenting a positive mood in their daily work, employees may convey enthusiasm and excitement, which engender a positive

affective response from their leaders. In return, such a response can guide leaders to positively evaluate and take on their employees' voices [22]. Previous laboratory [e.g., 23] and field studies [e.g., 24] have confirmed that such an affect-led process needs less cognitive effort. Therefore, we propose the following:

**H2:** Positive mood is positively associated with voice endorsement.

## The moderating role of felt obligation

The ELM also posits that one's position on the elaboration likelihood continuum is contingent upon one's ability and motivation. Our study indicates that an employee's voicer credibility or positive mood can affect voice endorsement depending on the felt obligation (motivation) and cognitive flexibility (ability) of the employee's leader to elaborate the merits of a focal topic [14, 25].

**Felt obligation.**   As the problems faced by organizations become increasingly complex and challenging to solve, leaders generally need to exert a high-level cognitive effort to elaborate upon issue-relevant information via the central route [26]. However, due to limited cognitive resources, they usually avoid participating in central routes unless the situation encourages them to do so [27]. Felt obligation for the organization (henceforth, felt obligation), which refers to an essential internal motivation that drives leaders to reciprocate prosocial behaviors with their organizations [28], is such a situation.

Building on the ELM literature, we predict that the positive relationship between voice credibility and voice endorsement becomes stronger when a leader's felt obligation is high. Specifically, employee credibility sends cues to the leader that the received voice is safe and worthy of endorsement. However, the leader may not necessarily respond to useful, credible suggestions unless that leader has a pro-organizational motivation to make the organization work more effectively [10]. Leaders with a low felt obligation are less committed to their organizations and thus usually put their own interests first. In that case, even if employees have demonstrated their expertise in previous work, their leaders may ignore it. As a result, a high felt obligation could increase the likelihood that leaders cognitively engage in the central route and simultaneously decrease their reliance upon the peripheral route characterized as less cognitive. Therefore, we propose the following:

**H3A:** High felt obligation strengthens the effect of voicer credibility on voice endorsement.

**H3B:** High felt obligation attenuates the effect of positive mood on voice endorsement.

## The moderating role of cognitive flexibility

**Cognitive flexibility.**   The ELM also posits that the likelihood of a person using the central or peripheral route in decision-making depends on ability [25]. In complex situations, individual decision-making is often limited by the human nature to participate in less cognitive processes, where existing rules of thumb or routinized schema are used to simplify the complicated aspects of an event. However, a leader with high cognitive flexibility can fight this natural tendency. Specifically, empowered by the awareness of other available options, a willingness to adapt to situations, and confidence in being flexible, the leader can maintain different representations of knowledge, information, and behavioral patterns in mental processes. As a result, the leader can think through multiple the alternatives offered by those employees with voicer credibility and choose the most effective response [29], leading to a stronger positive relationship between voicer credibility and voice endorsement.

In contrast, high cognitive flexibility attenuates the effect of positive mood on voice endorsement. As mentioned above, due to emotional contagion and the tendency towards less cognitive effort, positive employee mood could influence a leader's decision on whether to endorse the voice. However, this automatic process is less likely to occur when leaders have high cognitive flexibility because that flexibility facilitates the mental processes using mechanisms that are more cognitive rather than emotional for dealing with the problem at hand [30]. As the peripheral route is inhibited, the positive relationship between positive mood and voice endorsement is attenuated. Taken together, we hypothesize the following:

**H4A:** High cognitive flexibility strengthens the effect of voicer credibility on voice endorsement.

**H4B:** High cognitive flexibility attenuates the effect of positive mood on voice endorsement.

## Materials and methods

### Sample and procedures

The study was approved by the Academic Committee of Zhejiang University. All participants provided written informed consent when they were filling out the questionnaires. From April to June 2019, we contacted 200 team leaders (or department managers) working for 73 firms in Zhejiang, China. Survey data were collected mostly from cities of Wenzhou, Yiwu, and Taizhou, where the manufacturing industry was more developed (the data used in the present study were part of a broader data collection effort). We used a paired-questionnaire survey design. During the specific process of data collection, we usually invited all participants (4 to 6 participants per company in general) from the company to a nearby conference room, explained how to fill out the questionnaire with explicit instructions, and promised the confidentiality of all individual responses to reduce their worries about information leakage. For each matched pair of leader and employee, we would distribute two different envelopes, one containing the leader version of the questionnaire and the other containing the employee version of the questionnaire.

We then gave all participants adequate time to complete their questionnaires, which they put into the sealed envelopes thereafter. Each employee completed items related to his/her own voicer credibility, positive mood, and demographic information. Each employee's supervisor provided evaluations of his or her own voice endorsement, felt obligation, cognitive flexibility, and demographic information. In total, we sent out 400 questionnaires (200 for leaders and 200 for employees) and received responses from 173 leaders (86.5% response rate) and 182 employees (91.0% response rate)). After the deletion of invalid or unmatched questionnaires, we finally obtained 168 dyads. Based on previous studies [31, 32], a sample of 168 is sufficient to perform the regression analysis.

Table 1 shows some basic information of the sample. As indicated below, most participants who filled out the questionnaires were male. Specifically, the proportion of male employee participants is 64.3% and that of male leader participants is 74.4%. From the categories of organization tenure, we can find that the majority of employee participants have been working at their companies for less than 3 years (a proportion of 54.2%), while that of leader participants is for more than 3 years (a proportion of 72.1%). Regarding the education level, a large proportion of both employee (50.6%) and leader participants (34.5%) are in college, although the latter have a relatively higher percentage of higher education (more than college) than the former.

**Table 1. Background information of the employee and leader participants.**

| Demographic variables | Category | Frequency | Percentage |
|---|---|---|---|
| Employee's gender | Male | 108 | 64.3% |
| | Female | 60 | 35.7% |
| Employee's organization tenure | Within 1 year | 50 | 29.8% |
| | 1~3 years | 41 | 24.4% |
| | 3~8 years | 32 | 19.0% |
| | More than 8 years | 45 | 26.8% |
| Employee's education | middle school or below | 33 | 19.6% |
| | high school | 15 | 8.9% |
| | junior college | 21 | 12.5% |
| | college | 85 | 50.6% |
| | more than college | 14 | 8.3% |
| Leader's gender | Male | 125 | 74.4% |
| | Female | 43 | 25.6% |
| Leader's organization tenure | Within 1 year | 13 | 7.7% |
| | 1~3 years | 34 | 20.2% |
| | 3~8 years | 69 | 41.1% |
| | More than 8 years | 52 | 31.0% |
| Leader's education | middle school or below | 5 | 3.0% |
| | high school | 21 | 12.5% |
| | junior college | 58 | 34.5% |
| | college | 58 | 34.5% |
| | more than college | 26 | 15.5% |

## Measurement

All measures were administered in Chinese. We used a standard translation-back-translation procedure for any measures that were originally published in English to ensure validity [33].

Voice endorsement. To measure voice endorsement, we used a three-item managerial endorsement scale developed by Lam, Lee, and Sui (2019) [3]. Leaders were asked to provide ratings of voice endorsement towards the corresponding employees from 1 ("strongly disagree") to 5 ("strongly agree"). These items include: "This employee's suggestion has been, is being, or will be implemented," "I agree with this employee's comments," and "This employee's recommendation is valuable." The Cronbach's alpha was 0.87.

**Voicer credibility.** We adapted five items from Ohanian's (1990) source-credibility scale to develop a measure of voicer credibility [34]. Employees were asked to evaluate themselves on a five-point scale from 1 (strongly disagree) to 5 (strongly agree). These items include: "I am an expert in what I do," "I am experienced," "I am knowledgeable," "I am qualified," and "I am skilled." The Cronbach's alpha was 0.87.

**Positive mood.** According to Robinson and Clore (2002), when short time periods are employed, respondents mainly use the knowledge based on experience to report their state of mood, and when long time periods are employed, respondents mainly rely on the knowledge based on belief [35]. Given our concern with measuring positive mood as an experience rather than as a belief, we used the one-week time frame that was widely used in previous research [36] and asked employees to rate on a scale from 1 (very slightly or not at all) to 5 (extremely) about how they had felt at work in terms of enthusiastic, excited, happy, and delighted during the last week [19]. The Cronbach's alpha was 0.88.

**Felt obligation.** We adapted Eisenberger et al.'s (2001) 7-item felt obligation scale to measure a leader's desire to care about his or her organization and to help it reach its goals [37]. Leaders are asked to rate on a five-point scale from 1 (strongly disagree) to 5 (strongly agree). Sample items are "I would feel an obligation to take time from my schedule to help the organization if it needed my help." and "I have an obligation to the organization to ensure that I produce high-quality work." The Cronbach's alpha was 0.94.

**Cognitive flexibility.** We measured cognitive flexibility using a 12-item scale designed by Martin and Rubin (1995). This scale measures three components: awareness of communication alternatives, willingness to adapt to the situation, and self-efficacy in being flexible, each dimension containing four items [38]. Leaders are asked to rate on a five-point scale from 1 (strongly disagree) to 5 (strongly agree). Sample items are "I can find workable solutions to seemingly unsolvable problems" (awareness), "I am willing to work at creative solutions to problems" (willingness), and "I have the self-confidence necessary to try different ways of behaving" (self-efficacy). The Cronbach's alpha was 0.90.

**Control variables.** We controlled for three types of variables. First, we controlled for the variables associated with an employee's demographic characteristics that might influence the level of his or her leader's voice endorsement [19], including employees' gender, education level, and organizational tenure. Second, because a person's felt obligation or cognitive flexibility was related to his or her demographics [e.g., 39, 40], we controlled for leaders' demographic variables, including gender, education, and organization tenure. Third, we also controlled for voice frequency by asking employees to rate on a survey item from 1 (very few or not at all) to 4 (very often) in terms of how often they expressed voice to their leaders [3].

## Results

### Confirmatory factor analysis

To provide evidence of construct distinctness, we conducted confirmatory factor analysis (CFA) on the survey items from the five variables: voicer credibility, positive mood, felt obligation, cognitive flexibility, and voice endorsement. Using data obtained from 168 matched samples, we compared five alternative models with the baseline model, five-factor model 1. As shown in Table 2, the hypothesized five-factor structure of model 1 with all items loading on their respective factors fit the data in an acceptable way with $\chi^2$ [199, n = 165] = 372.97, RMSEA = 0.072, CFI = 0.94, IFI = 0.94, TLI = .93, and SRMR = 0.057 (Browne & Cudeck, 1993), which provided substantial improvement in fit indexes over the other alternative models (models 2–6). Additionally, all standardized factor loadings were above .40 and significant. These results suggest that the five constructs captured distinctiveness as expected in the study.

**Table 2. Comparison of measurement models.**

| Models | Factors | $\chi^2$ | df | $\chi^2$/df | RMSEA | CFI | IFI | TLI | SRMR |
|---|---|---|---|---|---|---|---|---|---|
| 1 | *Five factors*: Positive mood, voicer credibility, felt obligation, cognitive flexibility, voice endorsement | 372.97 | 199 | 1.87 | 0.072 | 0.94 | 0.94 | 0.93 | 0.057 |
| 2 | *Four factors*: Felt obligation and cognitive flexibility combined into one factor. | 560.26 | 203 | 2.75 | 0.103 | 0.88 | 0.87 | 0.85 | 0.088 |
| 3 | *Four factors*: Positive mood and voicer credibility combined into one factor. | 749.92 | 203 | 3.69 | 0.127 | 0.80 | 0.80 | 0.77 | 0.134 |
| 4 | Three factors: Positive mood and voicer credibility combined into one factor; felt obligation and cognitive flexibility combined into one factor. | 936.41 | 206 | 4.55 | 0.146 | 0.73 | 0.73 | 0.70 | 0.149 |
| 5 | *Two factors*: Supervisor ratings (e.g., cognitive flexibility) combined into one factor; subordinate ratings (e.g., positive mood) combined into one factor. | 1079.95 | 208 | 5.19 | 0.158 | 0.68 | 0.68 | 0.64 | 0.152 |
| 6 | *Single factor* | 1423.60 | 209 | 6.81 | 0.187 | 0.55 | 0.55 | 0.50 | 0.159 |

## Common method bias

We took several steps recommended by Podsakoff et al. (2003) to mitigate the influence of common method bias (CMB) [41]. First, because one of the major causes of CMB is obtaining the measures of variables from the same rater, we collected multi-source data using the paired-questionnaire technique. Second, we explained the research purpose in advance and ensured the anonymity of their responses. We told the respondents that there are no right or wrong answers to encourage them to provide honest responses. Third, we further carried out three different tests to verify that the CMB does not significantly influence the stability of our parameter estimates. (1) Harman's single factor test. The result shows that the variance for the first factor is 34.00% ($< 40\%$) [41], suggesting that CMB is not a major issue in the present study. (2) CFA approach. The CFA method in testing CMB considers that if the fit indicators of the single-factor CFA model does not meet the criteria of good fit, or if the single-factor CFA model is the worst-fitting model among the competing models, it means that the CMB is not serious [42]. As shown in Table 2, all indicators of the single-factor model ($\chi2$ /df = 6.81, RMSEA = 0.187, CFI = 0.55, IFI = 0.55, TLI = .50, and SRMR = 0.159) failed to meet the requirements and fit worse than the other five competing models. (3) ULMC Technique. To use the ULMC Technique, Richardson, Simmering, and Sturman (2009) suggested that researchers could first construct a method factor relating to all items in the hypothesized model. Then, researchers compare this new model with the hypothesized model [43]. If the variance of the two models is significantly different, then the CMB is severe; otherwise, the CMB is not severe. Following the procedures, we found that the variance between the two models is not significant [1.15, $\Delta\chi^2/\Delta df$ = (372.97–346.49)] / (199–176]], suggesting that the CMB is not severe in our study.

## Tests of convergent and discriminant validity

According to Ahmad et al. (2016), the convergent validity can be evaluated by the average variance extracted (AVE) and composite reliability (CR), while the discriminant validity can be evaluated by comparing AVE and the squared correlations involving the constructs [44]. As illustrated in Table 3, all the CR values are above 0.70, and the AVE values are above 0.5,

Table 3. Tests for convergent and discriminant validity.

| Variables | Convergent validity | Discriminant validity |
|---|---|---|
| **Voicer credibility** | 0.868 | AVE/(Corr)$^2$ > 1 |
| CR | 0.688 | |
| AVE | | |
| **Positive mood** | 0.886 | AVE/(Corr)$^2$ > 1 |
| CR | | |
| AVE | 0.661 | |
| **Felt obligation** | 0.949 | AVE/(Corr)$^2$ > 1 |
| CR | | |
| AVE | 0.729 | |
| **Cognitive flexibility** | 0.795 | AVE/(Corr)$^2$ > 1 |
| CR | | |
| AVE | 0.574 | |
| **Voice endorsement** | 0.868 | AVE/(Corr)$^2$ > 1 |
| CR | | |
| AVE | 0.688 | |

suggesting good convergent validity for all constructs. Moreover, the results of discriminant validity tests (AVE/(Corr)2 > 1) for all constructs show that the amount of the variance capture by each construct is greater than the shared variance with the other constructs, suggesting that the constructs are distinct from one another.

## Hypothesis testing

We employed ordinary least squares (OLS) regression analysis to evaluate the hypotheses. Before running the regression, we checked the linear regression assumptions in terms of multicollinearity, homoskedasticity, normality of the residual, and autocorrelation. The results showed that our data characteristics meet all the assumptions of the linear regression. (1) Multicollinearity. When multicollinearity is observed, the association between the variables leads to larger standard deviations and wide confidence intervals for the results. In our study, the maximum value of variance inflation factor (VIF) is 1.55 (<10), suggesting that the multicollinearity problem does not significantly influence the stability of the parameter estimates. (2) Homoskedasticity. Homoscedasticity is one of the essential assumptions of linear regression, which refers to the fact that the random error terms in the overall regression function have the same variance. In our study, we found that there is no obvious pattern in the distribution of the variance. (3) Normality of the residual. The third important assumption of linear regression is that the error term should obey a normal distribution. Otherwise, the confidence intervals for the estimated statistical results can become highly unstable. In our study, the P-P Plot plots fall approximately on a straight line, and the frequency of the regression standardized residual in the histogram presents a good normal distribution. (4) Autocorrelation. When autocorrelation occurs, the standard deviation measured tends to be smaller, leading to narrower confidence intervals. In our study, the Durbin-Watson (DW) value is 2.14 (close to the expected value of 2), and a further calculation showed that it is between $d_u$ and $4—d_u$, indicating no autocorrelation in our regression analysis.

The results are presented in Table 4. Model 1 includes only the dependent variable and control variables. Models 2 includes the dependent variable, the independent variable of voicer credibility, and the control variables. Model 3 includes the dependent variable, the independent variable of positive mood, and the control variables. Model 4 includes the dependent variable, the two independent variables, and the control variables. Model 5 includes the dependent variable, the two independent variables, the moderated variable of felt obligation and its moderating effect with voicer credibility, and the control variables. Model 6 includes the dependent variable, the two independent variables, the moderated variable of felt obligation and its moderating effect with positive mood, and the control variables. Model 7 includes the dependent variable, the two independent variables, the moderated variable of cognitive flexibility and its moderating effect with voicer credibility, and the control variables. Model 8 includes the dependent variable, the two independent variables, the moderated variable of cognitive flexibility and its moderating effect with positive mood, and the control variables.

Hypothesis 1 predicted that voicer credibility would have a positive impact on voice endorsement. As shown in Model 2, voicer credibility was statistically significant and positively correlated to voice endorsement ($b = .18$, $p < .01$). Thus, Hypotheses 1 was supported. Similarly, positive mood was found to affect voice endorsement positively as expected ($b = .17$, $p < .05$, see model 3), so Hypotheses 2 is also supported.

We then move on to test the moderating effects of felt obligation. Hypotheses 3A and 3B examine the impacts of voicer credibility and positive mood on voice endorsement under varied felt obligation conditions. We found that the interaction effect between felt obligation and voicer credibility was positive and significant for predicting voice endorsement ($b = .17$, $p <$

**Table 4. Results of hierarchical regression analyses.**

| Variable | Model 1 | Model 2 | Model 3 | Model 4 | Model 5 | Model 6 | Model 7 | Model 8 |
|---|---|---|---|---|---|---|---|---|
| **Control variables** | | | | | | | | |
| E's gender | −.08 (.12) | −.10 (.11) | −.11 (.12) | −.13 (.11) | −.11 (.09) | −.06 (.09) | −.02 (.09) | −.03 (.09) |
| E's educational | .12** (.04) | .13**(.04) | .10 *(.04) | .10 *(.04) | .06 (.04) | .04 (.03) | .06 (.04) | .05 (.04) |
| E's organizational tenure | .01 (.01) | .01 (.01) | .00 (.01) | .00 (.01) | −.00 (.01) | −.00 (.01) | .00 (.01) | .00 (.01) |
| L's gender | −.19 (.13) | −.17 (.13) | −.16 (.13) | −.14 (.13) | −.02 (.12) | −.02 (.11) | .03 (.11) | .02 (.11) |
| L's educational | −.03 (.06) | −.03 (.06) | −.04 (.01) | −.00 (.01) | .01 (.04) | .02 (.04) | .00 (.05) | .01 (.05) |
| L's organizational tenure | −.00 (.01) | −.00 (.01) | −.00 (.01) | .01 (.05) | .00 (.01) | −.00 (.01) | .00 (.01) | .00 (.01) |
| Voice frequency | −.01 (.05) | .01 (.05) | −.01 (.05) | | −.02 (.05) | −.00 (.04) | .01 (.04) | .01 (.04) |
| **Main effects** | | | | | | | | |
| Voicer credibility | | .18** (.07) | | .18** (.06) | .07 (.07) | .07 (.06) | .12 * (.05) | .09 (.06) |
| Positive mood | | | .17 * (.07) | .17 * (.07) | .03 (.06) | .15** (.06) | .14 * (.06) | .16** (.06) |
| Felt obligation | | | | | .49** (.07) | .48** (.06) | | |
| Cognitive flexibility | | | | | | | .65** (.08) | .63** (.08) |
| **Moderating effects** | | | | | | | | |
| Voicer credibility × Felt obligation | | | | | .17** (.07) | | | |
| Positive mood × Felt obligation | | | | | | −.30** (.06) | | |
| Voicer credibility × Cognitive flexibility | | | | | | | .11 (.09) | |
| Positive mood × Cognitive flexibility | | | | | | | | −.18* (.09) |
| $R^2$ | .07 | .11 | .10 | .15 | .42 | .49 | .42 | .43 |
| $\Delta R^2$ | | .04* | .03 * | .08* | .27** | .34** | .27** | .28** |

*Note.* n = 168 dyads. The standard errors in the estimations are reported in parentheses. Model 1 was the basis of comparison for Models 2, 3, 4, and Model 4 was the basis of comparison for Models 5, 6, 7, 8.

$* p < .05$,

$** p < .01$, two tailed tests.

.01, see model 5). In contrast, the interaction effect between felt obligation and positive mood was negative and significant ($b = −.30$, $p < .01$, see model 6). To explicate these interactions, we drew separate plots for leaders whose scores were one standard deviation below and above the mean of felt obligation (Cohen & Cohen, 1983). As illustrated in Fig 2, the relationship between felt obligation and voicer credibility was positive and significant for the group with higher felt (+1SD, $b = .21$, $p < .05$), but this relationship was negative and nonsignificant for the group with lower felt obligation (−1SD, $b = −.07$, n.s.). A different pattern of results was found for positive mood. As illustrated in Fig 3, the relationship between positive mood and voice endorsement was positive and significant for the group with lower felt obligation (−1SD, $b = .38$, $p < .01$) but negative and nonsignificant for the group with higher felt obligation (+1SD, $b = −.08$, n.s.). These results show that the moderating effect of felt obligation was positive and significant in the central route, and was negative and significant in the peripheral route. Thus, Hypothesis 3A and 3B are both supported.

The procedure for testing the moderating effects of cognitive flexibility was identical. Hypotheses 4A and 4B examine the impacts of voicer credibility and positive mood on voice endorsement under varied cognitive flexibility conditions. Hypotheses 4A predicts that the positive relationship between voicer credibility and voice endorsement will be more pronounced when leaders have more cognitive flexibility. However, we found that although the voicer credibility × cognitive flexibility term was positive, it was statistically insignificant ($b = .11$, n.s., see Model 7). Thus, Hypothesis 4A was not supported. We then tested positive mood's interaction with cognitive flexibility, and the evidence showed that the positive

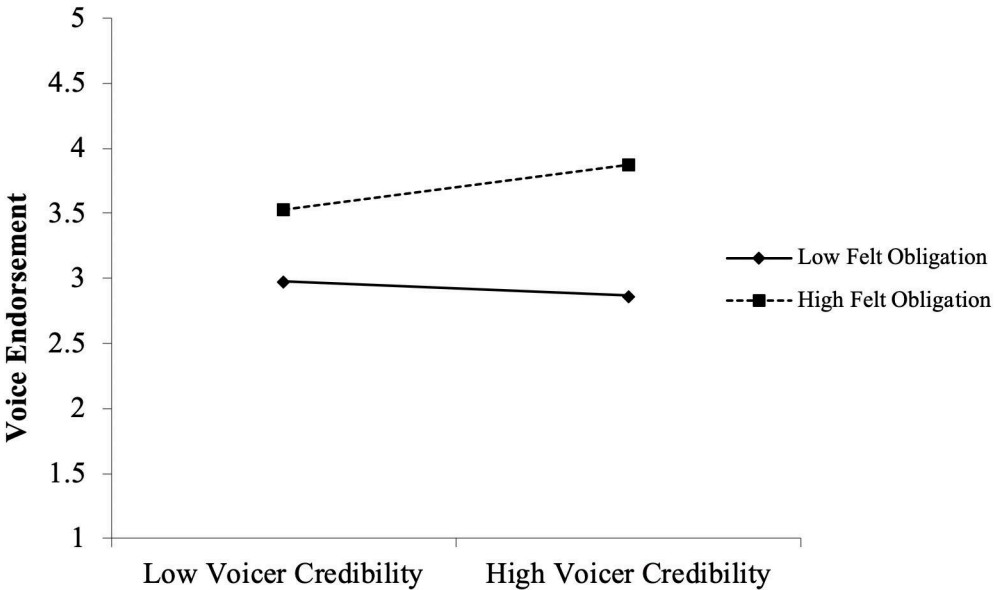

**Fig 2. Interactive effect of voicer credibility and felt obligation on voice endorsement.**

relationship between positive mood and voice endorsement was negative and significant ($b = -.18$, $p < 0.05$, see Model 8), Hypothesis 4B thus is supported. A plot of this interaction shown in Fig 4 further indicated that the relationship between cognitive flexibility and positive mood was nonsignificant for the group with higher cognitive flexibility (+1SD, $b = .04$, n.s.), and was positive and significant for the group with lower cognitive flexibility (−1SD, $b = .29$, $p < .01$). These results show that the moderating effect of cognitive flexibility was negative and significant in the peripheral route, but was not significant in the central route, thus lending support for Hypothesis 4B but not for Hypothesis 4A.

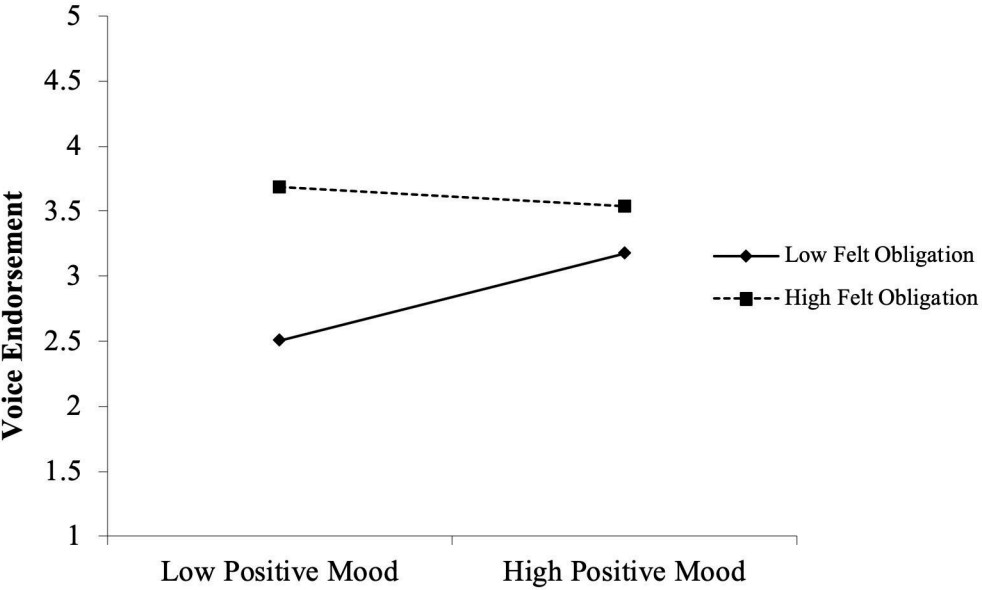

**Fig 3. Interactive effect of positive mood and felt obligation on voice endorsement.**

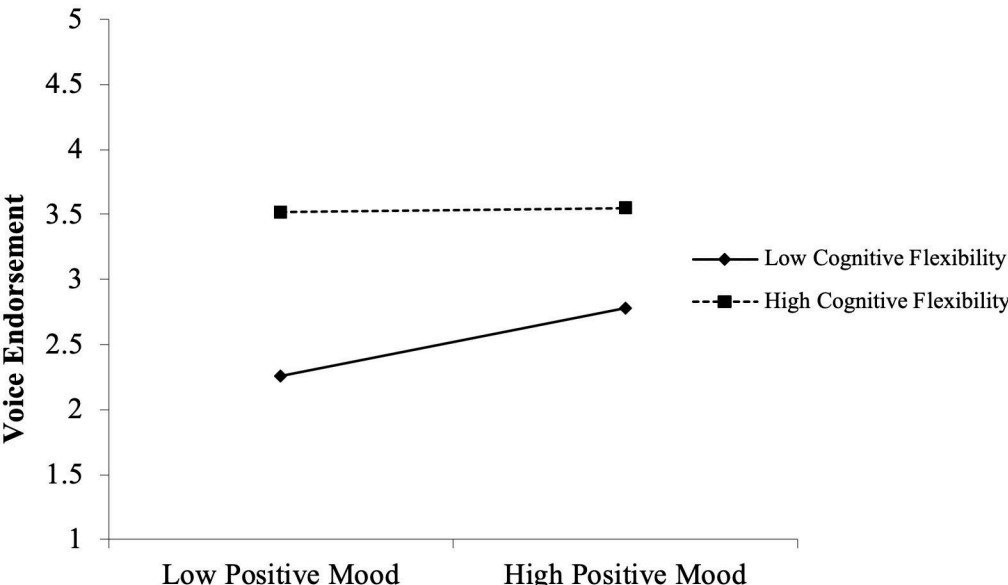

**Fig 4. Interactive effect of positive mood and cognitive flexibility on voice endorsement.**

## Discussion

In this study, we leveraged ELM to provide a theory for how leaders are persuaded by their employees. Synthesizing what we observed from the results, we found considerable support for ELM in voice endorsement. Specifically, we found that the sender factors, i.e., issue-relevant information from employees about voicer credibility and peripheral cues of positive mood, are both positively related to voice endorsement. The results also indicated that the receiver factors, i.e., the felt obligation and cognitive flexibility of leaders, play a vital role in the above relationships. For motivation, we found that higher felt obligation strengthens the positive effect of voicer credibility on voice endorsement but attenuates the positive effect of positive mood on voice endorsement. For ability, we found that higher cognitive flexibility attenuates the positive effect of positive mood on voice endorsement but has no significant impact on voicer credibility and voice endorsement.

We believe that the result of Hypothesis 4 is not valid for two reasons. First, we found that the majority of our sample came from traditional industries such as manufacturing, where the environment faced by leaders is generally predictable, and the problems to be solved are mostly well structured. In such a context, we speculate that either the high voice credibility from employees or the high cognitive flexibility from leaders is sufficient for leaders to engage in the cognitive processes needed. Second, this non-significance may be due to the broad fact that the interaction between voicer credibility and cognitive flexibility is a substitutive rather than a synergic process. Leaders with high cognitive flexibility may be overconfident and ignore the benefits of employee voicer credibility to the organization. Similarly, when faced with highly credible employees, leaders may rely too heavily on those employees, leading to a decreased willingness to participate in the cognitive processes needed.

### Theoretical and practical contributions

We make several key theoretical contributions. First, we demonstrate that employees can use their mood to affect leader decision-making regarding voice endorsement. This result

highlights the importance of positive mood in voice endorsement, which echoes Morrison's (2011) call that further theory building and empirical work are needed to reveal the role of mood in voice research [45]. Prior work has begun to explore the importance of employee mood in the communication process of voice. However, most of these studies assumed that the transmission of mood was a one-way process in which high-power individuals influence low-power individuals and thus did not realize the importance of employee mood in leaders' decision-making behaviors. That is certainly plausible because in employee voice (upward communication), employees ascertain the favorability of the social context for voice by observing the mood of their leaders [19]. However, with voice endorsement (downward communication) attracting research interests from scholars, a question worthy of investigation is, "Will employees who exhibit a positive mood influence voice endorsement by their leaders?" Unlike previous studies that largely examined how high-power leader moods influence lower-power employee voice [e.g., 19], the present study provides evidence for the reverse process in which lower-power employees can affect high-power leader voice endorsement through positive mood. As such, we add to the voice literature in two main ways: by introducing positive mood as a determinant of voice endorsement and by recognizing the importance of employee mood in downward (voice) communication.

Second, our findings contribute to and extend the voice literature by introducing the social persuasion theory of ELM into voice endorsement. Prior studies on voice endorsement, although limited, are quite diverse and have investigated a variety of determinants from aspects of the voice message, sender, and receiver [4]. For example, scholars have sought to explore the influence of message-factors on voice endorsement, such as issue importance [8]. Alternatively, others have shown the impact of sender factors such as trustworthiness [17] and receiver factors such as managerial self-efficacy [46]. However, if we scrutinize these determinants with ELM, most of them can be grouped into issue-relevant information, suggesting that the determinants associated with peripheral cues are under-investigated. The present study fills this gap by proposing a dual-path structure of ELM to investigate voice endorsement, including issue-relevant information (i.e., voicer credibility) and a peripheral cue (i.e., positive mood). The results also indicate that although both voicer credibility and positive mood have a positive impact on voice endorsement, they do so through two different paths—the central and peripheral routes. By distinguishing and integrating the determinants of voice endorsement from the ELM perspective, we inform a new way of theoretically encompassing the existing and unexamined determinants of voice endorsement and thereby opening up new avenues for future research.

Third, taking the ELM perspective also contributes to the voice literature by providing a holistic manner to examine how the interplay of sender factors and receiver factors affects voice endorsement. As suggested by Schreurs et al. (2020), it is important to understand the phenomenon of voice endorsement in a comprehensive way because voice communication is complicated and dynamic and because leaders are likely to multitask several cues in a voice event. Unlike previous work on voice endorsement that independently examined the influence of sender and receiver factors, the present study uses the ELM perspective to explicitly account for voice endorsement by studying the interactive effects of sender factors (voicer credibility and positive mood) and receiver factors (felt obligation and cognitive flexibility) in a single study.

Our findings also offer valuable practical implications for both employees and managers. From the employee perspective, it will be important to understand how they present themselves will affect their leaders' cognitive or affective processes of voice endorsement. Building credibility often takes a lengthy period and many work-based interactions, and it is difficult for employees to change their leaders' perception of trust in them within a short time.

Therefore, one possible action strategy they can employ is to present positive emotions towards their work, colleagues, and supervisors if they expect their voice to be taken. Meantime, our results also suggested that leaders' motivation and ability are important moderators of the associations between voicer credibility (positive mood) and managerial endorsement, so employees should get to know their leaders before speaking up. For example, if an employee notices that his or her leader is a person who can handle complicated issues but lacks a sense of obligation to the organization, they must be aware that showing a positive mood may have little impact on their leader's voice endorsement. The employee should be cognizant of their credibility before speaking up. Especially for those less credible employees, it may be better to build their credibility first. From the manager or organizational perspective, they usually expect their decision of voice endorsement to be objective and fact-based rather than being driven by emotions. In this case, the organization should increase managers' sense of responsibility for the organization or their cognitive ability to address complex issues through management efforts, such as carrying out regular communication meetings on organizational goals or theoretical training activities that combine company practices.

### Limitations and future research

This study has several limitations. First, our study was conducted in China, which may limit the generalization of our results to other culturally different contexts. In high-context societies such as China, individuals are more likely to focus on (and make use of) mood because their main goal in interactions is to preserve harmony and save face for others [47]. Meanwhile, individuals in high-context cultures rely more on shared understandings (contexts) to convey information, so they tend to be more comfortable with ambiguous messages and less emphasis on the content of the information being exchanged [47]. Thus, compared to respondents from low-context cultures, the leader respondents in the present study may be more sensitive to employee mood and less sensitive to employee credibility, resulting in the impact of positive mood being magnified and the impact of voicer credibility being reduced. Therefore, future research can be conducted in low-context cultures to examine whether our findings remain valid or even introduce the low-/high-context cultures as a moderating variable to explore its impact on voice endorsement. Moreover, in explaining why the moderation of cognitive flexibility is not significant, we found that our samples were largely from manufacturing, which may cause a generalization problem. Therefore, future research could test our findings using data from different industries and cultures to improve the generalizability of our findings.

Second, although we advance the voice literature by focusing on the interplay of the sender and receiver factors, we may have missed the opportunity to understand the phenomena of voice endorsement more explicitly due to the exclusion of message factors from our dual-path model. As indicated by Schreurs et al. (2020), voice endorsement is an interactive function of the message with sender and receiver factors. Future research could, therefore, pull together all these factors in a single study. In addition, we encourage research to explore other determinants of voice endorsement based on the perspective of ELM. For example, we examined only positive mood as a peripheral cue in our study. However, as another peripheral cue, negative employee mood could also influence leader voice endorsement because it can trigger affective responses such as defense and fear [19]. Besides employee mood, other peripheral cues such as employees' body gestures, linguistic traits, and the combinations (selections) of the words may also influence the voice endorsement of their leaders and is worthy of further investigation.

Third, despite our efforts to improve the reliability of our results, we still use self-reported data. Because respondents tend to beautify their mood or exaggerate their abilities, we may have produced a social desirability problem that led to biased results in this study. Although it

is common to use self-report scales such as mood [48] and cognitive ability [49] in voice research, we do recognize the limitations of self-reported measures and call for future research using peer-assessment or objective techniques (e.g., fMRI and brain scanning) to validate our findings.

## Supporting information

**S1 Data.**
(SAV)

## Author Contributions

**Conceptualization:** Xiaobo Li, Ting Wu.

**Formal analysis:** Xiaobo Li, Ting Wu.

**Funding acquisition:** Ting Wu.

**Methodology:** Ting Wu, Jianhong Ma.

**Supervision:** Ting Wu.

**Validation:** Jianhong Ma.

**Writing – original draft:** Xiaobo Li.

**Writing – review & editing:** Ting Wu, Jianhong Ma.

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
