## [Decision Letter · Decision Letter 0]

26 Feb 2021

PONE-D-20-29327

How leaders are persuaded: an elaboration likelihood model of voice endorsement

PLOS ONE

Dear Dr. WU,

Thank you for submitting your manuscript to PLOS ONE. After careful consideration, we feel that it has merit but does not fully meet PLOS ONE’s publication criteria as it currently stands. Therefore, we invite you to submit a revised version of the manuscript that addresses the points raised during the review process.

In the revised version of the paper, please address the reviewers' comments listed at the bottom of this email and try to better present the statistical analysis performed in the paper.

We look forward to receiving your revised manuscript.

Kind regards,

Camelia Delcea

Academic Editor

PLOS ONE

Journal Requirements:

"This work was supported by Hangzhou Social Science Project for Developing High

Calibre Youth Talent under Grant No. 2018RCZX22, Zhejiang Provincial Natural

Science Foundation of China under Grant No. LQ18G020007."

" NO - Include this sentence at the end of your statement: The funders had no role in study design, data collection and analysis, decision to publish, or preparation of the manuscript."

Reviewers' comments:

Reviewer's Responses to Questions

**Comments to the Author**

1. Is the manuscript technically sound, and do the data support the conclusions?

Reviewer #1: Yes

Reviewer #2: Yes

Reviewer #3: Yes

Reviewer #4: Partly

2. Has the statistical analysis been performed appropriately and rigorously? 

Reviewer #1: Yes

Reviewer #2: Yes

Reviewer #3: Yes

Reviewer #4: No

3. Have the authors made all data underlying the findings in their manuscript fully available?

Reviewer #1: No

Reviewer #2: Yes

Reviewer #3: Yes

Reviewer #4: No

4. Is the manuscript presented in an intelligible fashion and written in standard English?

Reviewer #1: Yes

Reviewer #2: Yes

Reviewer #3: Yes

Reviewer #4: Yes

5. Review Comments to the Author

Reviewer #1: The topic of voice endorsement is very relevant and apt. While the research on voice has been increasing rapidly, the focus on voice endorsement has been relatively sparse. This paper focuses on a very relevant theme and provides insights into the dynamics of voice in organizations. This was an interesting paper and I thoroughly enjoyed reading the paper. The paper is well written and clearly focuses on exploring the context of voice endorsement. The authors leverage the Elaboration Likelihood Model (ELM) to integrate and validate the various aspects related to voice endorsement. The paper has many strengths and is well written. However, I would like to share a few suggestions and concerns that may help improve the paper. The same are attached.

Reviewer #2: The research was conducted well, and it is a unique approach to understand about how voice endorsement can influence other people.

There are some things that needs to be improved upon:

1. You will need to defend why 168 paired samples is enough? Does it represents the overall populations?

2. You will also needs to inform about the data collection process in the study? How long? How did you get those data?

3. You will need to defend on why you use odd number of Likert Scale. Some researchers like presented in the study of Dhar and Simonson (2003), encourage even number of Likert scale to avoid middle answers.

4. Is there any practical contributions that can be presented from this study? What are the benefits for the businesses and practitioners of your work?

5. Related with the practical contributions, it can also be explored on the combination of not just voice alone, but perhaps with other traits such as body gestures, linguistic traits, selections the combinations of the words, and other factors.

Reviewer #3: Quiet an interesting study. The authors have explored a very unique area and the arguments and conclusions are well noted. The limitations and theoretical contributions are well noted. The study has more room for further research on this area and scope.

Reviewer #4: Dear Editor,

I want to thank the authors for their effort in providing this manuscript titled "How leaders are persuaded: an elaboration likelihood model of voice endorsement." I reviewed the manuscript with great interest as I feel this is an exciting area of research. In their manuscript, the authors provided an excellent background and a comprehensive review of the literature. In particular, I think the way they approached each hypothesis was easy to follow and to get a good idea about gaps in literature they want to cover. However, I find myself disappointed with the statistical analysis they provided, considering the effort they show in this manuscript.

1- In the section "common method bias," the authors described their ex-ante steps taken to minimize bias per Podesakoff et al. (2003).[1] Additionally, they mentioned using "commonly used" Harman's single factor test as an ex-post remedy. However, several publications and simulations have provided evidence against using such a test.[2,3] Therefore, the authors' confirmation about the absence of CMB in their research can be questionable if they are using this test alone to provide such a sense of security.

2- Table 3, from a statistical point of view, I am not convinced that producing mean, sd, and correlations for categorical data is sensible. I understand that many papers have been published with such tables, but I do not find such tables provide either correct information or have added value. I believe this table should not be included in the analysis in its current form. Frequencies, percentages, tests of associations, even if we want to use correlations as mentioned in the table, that type of correlation should be indicated.[4]

3- The authors used CFA and path modeling to explain the causal relationship in their analysis and provide evidence for their hypothesis testing. However, they are not providing a graphical representation of these relationships. They shifted to OLS and hierarchical regression analyses. Such a method is a big assumption from their side that the response variable is continuous. There is no measuring for the suitability of using OLS in the framework of their analysis. The authors did not check for OLS assumptions.

4- The concepts the authors are measuring are derived from latent variables that cannot be measured using the methods suggested in their analysis approach. Methods such as SEM and IRT should have been explored in the context of questionnaire data analysis. Moreover, I am not sure what is the response variable in their analysis using OLS. How the authors pooled the results into a single response variable?!

Conclusion

I find the data's analysis inadequate to the research and the type of data the authors provided. Therefore, I feel that this manuscript in it is current form is not suitable for publication. The data should be re-analyzed using more statistically sound approaches.

Bibliography

[1] Podsakoff PM, MacKenzie SB, Lee JY, Podsakoff NP. Common method biases in behavioral research: a critical review of the literature and recommended remedies. Journal of applied psychology. 2003 Oct;88(5):879.

[2] Aguirre-Urreta MI, Hu J. Detecting common method bias: Performance of the Harman's single-factor test. ACM SIGMIS Database: the DATABASE for Advances in Information Systems. 2019 May 6;50(2):45-70.

[3] Richardson HA, Simmering MJ, Sturman MC. A tale of three perspectives: Examining post hoc statistical techniques for detection and correction of common method variance. Organizational Research Methods. 2009 Oct;12(4):762-800.

[4] Khamis H. Measures of association: how to choose?. Journal of Diagnostic Medical Sonography. 2008 May;24(3):155-62.

6. PLOS authors have the option to publish the peer review history of their article (what does this mean?). If published, this will include your full peer review and any attached files.

Reviewer #1: No

Reviewer #2: **Yes: **Raden Aswin Rahadi

Reviewer #3: No

Reviewer #4: No

---

## [Author Response · Author response to Decision Letter 0]

15 Apr 2021

We have provided the responses to the academic editor in the cover letter, and responses to the reviewers in the file "Response to Reviewers”. In these two documents, the responses are presented more clearly.

---

## [Decision Letter · Decision Letter 1]

5 May 2021

How leaders are persuaded: an elaboration likelihood model of voice endorsement

PONE-D-20-29327R1

Dear Dr. WU,

We’re pleased to inform you that your manuscript has been judged scientifically suitable for publication and will be formally accepted for publication once it meets all outstanding technical requirements.

Kind regards,

Camelia Delcea

Academic Editor

PLOS ONE

Additional Editor Comments (optional):

Reviewers' comments:

Reviewer's Responses to Questions

**Comments to the Author**

1. If the authors have adequately addressed your comments raised in a previous round of review and you feel that this manuscript is now acceptable for publication, you may indicate that here to bypass the “Comments to the Author” section, enter your conflict of interest statement in the “Confidential to Editor” section, and submit your "Accept" recommendation.

Reviewer #2: All comments have been addressed

Reviewer #4: All comments have been addressed

2. Is the manuscript technically sound, and do the data support the conclusions?

Reviewer #2: Yes

Reviewer #4: Yes

3. Has the statistical analysis been performed appropriately and rigorously? 

Reviewer #2: Yes

Reviewer #4: Yes

4. Have the authors made all data underlying the findings in their manuscript fully available?

Reviewer #2: Yes

Reviewer #4: No

5. Is the manuscript presented in an intelligible fashion and written in standard English?

Reviewer #2: Yes

Reviewer #4: Yes

6. Review Comments to the Author

Reviewer #2: In overall I am satisfied with the authors response to my comments and suggestion for revisions. I am happy to accept their revision in their current form.

Reviewer #4: The authors have addressed all the points raised in the review. I was skeptical about their ability to provide the justification for many of the points raised in my comments. While I am still inclined against using OLS and hierarchical regression, they provided enough evidence to support acceptable results using this method. Therefore, I have no issues left against the manuscript.

7. PLOS authors have the option to publish the peer review history of their article (what does this mean?). If published, this will include your full peer review and any attached files.

Reviewer #2: **Yes: **Raden Aswin Rahadi

Reviewer #4: No

---

## [Editor Report · Acceptance letter]

7 May 2021

PONE-D-20-29327R1 

How leaders are persuaded: an elaboration likelihood model of voice endorsement 

Dear Dr. Wu:

I'm pleased to inform you that your manuscript has been deemed suitable for publication in PLOS ONE. Congratulations! Your manuscript is now with our production department. 

Kind regards, 

on behalf of

Dr. Camelia Delcea 

Academic Editor

PLOS ONE